# Taste of time: A porous-medium model for human tongue surface with implications for early taste perception

**Zhenxing Wu**, **Kai Zhao***

Department of Otolaryngology - Head & Neck Surgery, the Ohio State University, Columbus, Ohio, United States of America

* zhao.1949@osu.edu

**Data Availability Statement:** All relevant data are within the manuscript and its Supporting Information files.

**Funding:** This work was supported by funding from NIH NIDCD R01 DC013626 and R21

## Abstract

Most sensory systems are remarkable in their temporal precision, reflected in such phrases as "a flash of light" or "a twig snap". Yet taste is complicated by the transport processes of stimuli through the papilla matrix to reach taste receptors, processes that are poorly understood. We computationally modeled the surface of the human tongue as a microfiber porous medium and found that time-concentration profiles within the papilla zone rise with significant delay that well match experimental ratings of perceived taste intensity to a range of sweet and salty stimuli for both rapid pulses and longer sip-and-hold exposures. Diffusivity of these taste stimuli, determined mostly by molecular size, correlates greatly with time and slope to reach peak intensity: smaller molecular size may lead to quicker taste perception. Our study demonstrates the novelty of modeling the human tongue as a porous material to drastically simplify computational approaches and that peripheral transport processes may significantly affect the temporal profile of taste perception, at least to sweet and salty compounds.

## Author summary

Taste perception is an important gateway for food selection, food intake, energy and nutrition balance–as world is facing epidemic of obesity and diabetes. Information conveyed via the taste system provide crucial behavior choices, e.g. in identifying edible and nutritious food source, driving hedonic evaluation and craving, as well as avoiding poisonous substances. Thus, the interest to understand early taste responses is important, not only for basic science, but also for clinical and public health applications.

## Introduction

Temporal precision in sensory perceptions is important to their physiological functions—it is how we can see and respond to oncoming traffic or echo-locate a sound source. Yet for taste, its early temporal response is complicated by the transport and diffusion processes of the

DC017530 to KZ. The funders had no role in study design, data collection and analysis, decision to publish, or preparation of the manuscript.

**Competing interests:** The authors have declared that no competing interests exist.

stimuli on the surface of the tongue. The tongue's surface is covered by four different types of papillae (see Fig 1A): the circumvallate papillae forming an inverted V shape in the posterior one-third of the tongue, the foliate papillae in folds on the lateral borders of the tongue, and fungiform papillae scattering in the anterior two-thirds of tongue[1]. These three types papillae contain the taste buds[2] that are mainly responsible for gustatory perception. The fourth type, filiform papillae, are the most numerous; they do not contain taste buds and mainly serve to increase friction between the tongue and food or to transmit temperature and mechanical information[3]. These papillae combine to form a dense "kelp forest" (Fig 1B). Yet before taste perception can occur, taste stimuli must be transported via convection and diffusion through papilla gaps, saliva, taste pores, and so forth, to reach the taste receptors located within the taste buds on the papillae. Relatively little is known about these early taste events due to incomplete understanding of this complicated transport process. For example, there is a common agreement that, if we drop a taste solution on the tongue, taste perception (e.g., intensity) will increase with time, but we don't know whether it is because the neural taste system accumulates more "information" over time or because the stimulus concentration within the papillae rises with time.

This lack of understanding is surprising given that taste, one of the primary sensory systems, is an important gateway for our food selection, food intake, and energy and nutrition balance. Information conveyed via the taste system provides crucial behavior cues in

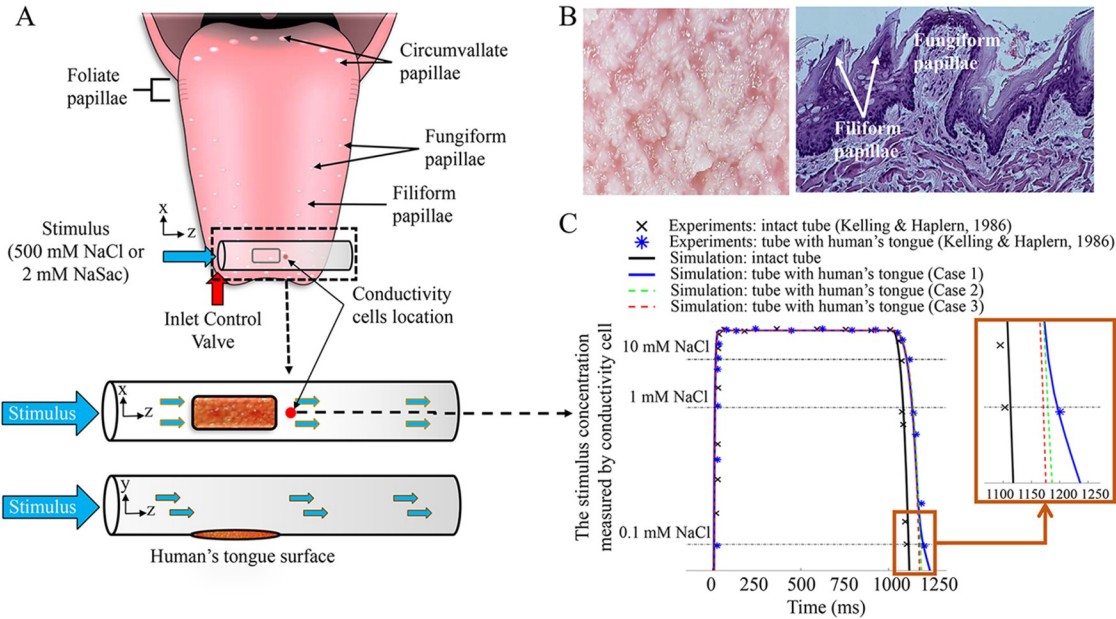

**Fig 1. *In silico* replication of Kelling and Halpern's experiment. (A)** Schematics of the experiment, where they placed a tube with a small opening across the top surface of the subject's tongue. Precise pulses (from 100 to 1000ms) were then delivered through the tube at 10 ml/s. (The tongue image is reproduced from https://commons.wikimedia.org/wiki/File:Human_tongue_taste_papillae. svg) **(B)** The filiform papillae and fungiform at anterior dorsal surface of the tongue and a histology section (images are reproduced from https://commons.wikimedia.org/wiki/File:%D0%AF%D0%B7%D1%8B%D0%BA_%D1%81%D0%BE%D1%81%D0%BE% D1%87%D0%BA%D0%B8.jpg and[4], Creative Common license) **(C)** Measurement of stimulus concentration using conductivity cells with and without tongue surface. Without the tongue surface (the tube is intact without an opening), the stimulus has an almost perfect square pulse (1000ms of 500Mm NaCl). With an opening covered by tongue surface, the pulse has significant delay during offset (inset). The simulation well matched experimental data.

identifying edible and nutritious food sources, driving hedonic evaluation and craving, and avoiding poisonous substances. Thus, understanding early taste responses is important not only for basic science but also for clinical and public health applications in a world facing an epidemic of obesity and diabetes.

In this study, we addressed this knowledge gap by modeling the transport processes of a range of sweet and salty stimuli within the papilla environment of the human tongue using a novel porous-medium approach. Based on electron microscopic scans of human tongue in the literature, we estimated the porosity and permeability of the papilla structure by well-established theoretical formulas. We then replicated the classical experiments conducted by Kelling and Halpern[5,6] as well as various sip-and-hold experiments[7–9] to understand how much the stimulus temporal transport profile may contribute to the taste perception profile.

## Results

### Modeling human tongue surface as a porous medium

Porous media are materials containing pores (or voids) that can take various shapes, forms (regular or irregular), and stiffness (e.g., sponge, wool, fibers, and rough surfaces) and have been heavily researched in traditional mechanical and industrial engineering[10–12]. Modeling fluid transport exactly in every fiber or every pore of a porous material (e.g., water soaking into a sponge) is highly complex and almost impossible; thus, the common approach is to model it based on key material properties[13–15]. Two of the most important properties are *porosity*, the fraction of the void (i.e., "empty") spaces in a material, between 0 and 1, or between 0% and 100%; and *permeability*, the ability of a porous material to allow fluids to pass through it.

We determined these material properties for the tongue's surface based on well-established theoretical formulas[12,16,17] for microfiber porous surfaces (Fig 2A; also in methods). Two key parameters of the papilla structure, distribution density ($D_p$) and average diameter ($d_p$), were estimated from reported human tongue morphology[18–26] (see methods). Since these parameters contain significant variation in the literature, we considered different sets of values to cover the ranges (Table 1) and validated them against experimental data.

### Early taste perception with rapid, precise pulses

We replicated *in silico* the classic temporal taste stimulus delivery experiments by Kelling and Halpern[5,6], who placed a tube with a small opening across the top surface of a subject's tongue (see Fig 1A, bottom). Precise pulses (from 100 to 1000ms) of 500mM sodium chloride (NaCl) or 2mM sodium saccharin (NaSac) were delivered through the tube at 10 ml/s, after which the subject rated perceived taste intensity of salty or sweet, respectively. In our computational model, we covered the opening of a tube of the same dimensions with a porous tongue surface (Fig 2A). We then simulated the pulses at the same flow rate as in the experiments, with and without an opening (Fig 1C). Without an opening, the stimulus has an almost perfect square pulse. With the tongue surface covering an opening, the pulse has a significant delay during offset, potentially due to the tongue surface acting as a porous "sponge" that traps some of the stimulus and affects the concentration profile downstream at the measurement site (see Fig 1A, bottom: red dot). One set of porosity and permeability values (Table 1, case 1) from our estimation most accurately matched the concentration measured experimentally by conductive cells, a confirmation that our estimation of porosity and permeability is valid. The benefit of using the porous-material approach is that, even though we estimated the porosity and permeability based on an idealized structure (Fig 2A), it is applicable to irregular and

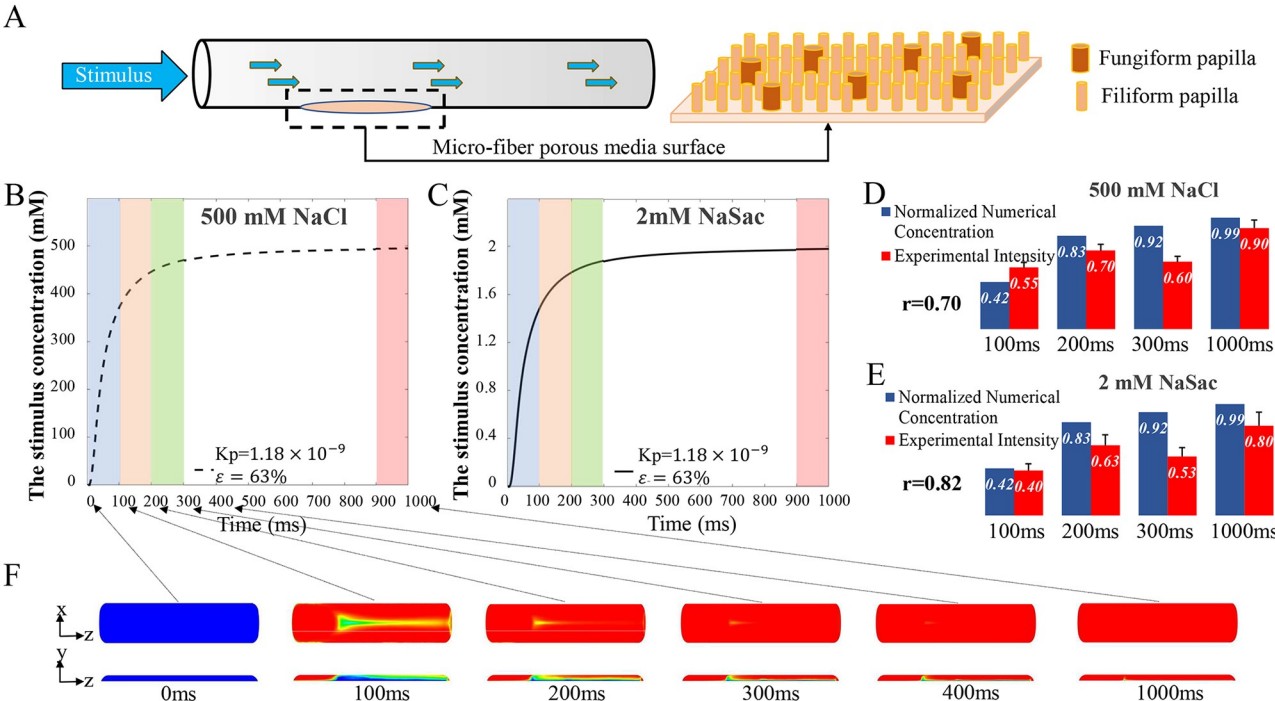

**Fig 2. Simulation of stimuli transport in tongue surface. (A)** Schematics of modeling papilla structure as microfiber porous media. **(B, C)** Averaged stimulus concentration within the exposed tongue surface zone as a function of time for 500mM NaCl **(B)** and 2mM NaSac **(C)** in a 1000ms pulse. **(D, E)** Simulated tongue surface stimulus concentration profile against the experimental perceived taste intensity for different pulses. Both data are normalized to 2s-long pulse. **(F)** Top-surface view (top row) and longitudinal cross-section view (bottom) of NaCl concentration profile of the papilla layer at different time points. (Also see S1 Video).

heterogeneous shapes and sizes of pores, as long as the material's porosity and permeability can be accurately validated through experiments.

Finally, we compared our simulated tongue-surface stimulus-concentration profile against the experimental perceived taste intensities[6]. The averaged stimulus concentration within the exposed tongue surface region rises with significant delay to an exponential-like curve for both NaCl and NaSac solutions (Fig 2B and 2C; see also S1 Video). Fig 2F shows that the top-surface NaCl concentration (top row) reaches close to saturation at ~300ms, whereas the cross-section view through the papilla layer (bottom row) shows significantly more delays at deeper layers. This indicates that the porous structure of human tongue surface does reduce and delay the effective stimulus-concentration buildup. Higher porosity and permeability cases (see S1 Fig) would increase the transport speed and effective concentrations.

**Table 1. Papilla structural properties and corresponding porous medium properties.**

| Case | Filiform papillae | | Fungiform papillae | | Total papilla density,$D_p$ (papilla/mm$^2$) | Averaged papilla diameter,$d_p$ (mm) | Porosity, $\varepsilon$ | Permeability, $K_p$ (m$^2$) |
|---|---|---|---|---|---|---|---|---|
| | Density (papilla/mm$^2$) | Diameter (mm) | Density (papilla/mm$^2$) | Diameter (mm) | | | | |
| Case 1 | 7 | 0.2 | 2 | 0.4 | 9 | 0.250 | 0.63 | $1.18\times10^{-9}$ |
| Case 2 | 7 | 0.15 | 1 | 0.55 | 8 | 0.200 | 0.79 | $2.78\times10^{-9}$ |
| Case 3 | 6 | 0.1 | 2 | 0.2 | 8 | 0.125 | 0.92 | $7.05\times10^{-9}$ |

In Kelling and Halpern's experiments, a 2-second pulse was used as a standard against which subjects would judge in proportion the taste intensity of the last 100ms of the experimental pulses, (e.g., from 100ms to 200ms for a 200ms pulse; from 200ms to 300ms for a 300ms pulse, etc.). The experimental results were shown in Fig 2D and 2E, red bar and S1 Table. Therefore, we averaged the concentration within the tongue surface region from 0 to 100ms, 100ms to 200ms, 200ms to 300ms, and 900ms to 1000ms for each pulse (see Fig 2B and 2C, colored bands), and normalized against a long 2-second pulse—the latter resulted in full saturation. Our simulation results match quite well with experimental data. For example, the average concentration of 500mM NaCl stimulus reaches only 42% of the saturated concentration with a 100ms pulse and 83% with a 200ms pulse (see Fig 2D). This is consistent with the experimental results: subjects rated the 100ms pulse of NaCl as only 55% as strong as a 2s saturating pulse, and the 200ms pulse as 70% of the saturating pulse[6]. The concentration-time profile for 2mM NaSac also matches well with experimental data, especially for the shorter pulses. For longer pulses, we suspect sensory adaptation might take effect, which our model does not address. Overall, the correlation between the simulated concentration and the experimental intensity results for all NaCl and NaSac stimulus pulses were 0.70 and 0.82 (see Fig 2D and 2E, and S1 Table), respectively.

### Taste perception during sip and hold

Next, we modeled traditional sip-and-hold experiments[7–9,27–29], where a subject takes a sip of a taste solution, holds it in the mouth for a given time period, and then rates perceived taste intensity. We assume there is no significant fluid movement during the sip and hold; thus, the stimulus penetration into the papilla structure is governed predominately by the diffusion process. The effective diffusivity of the stimuli in the porous material is estimated based on its porosity and porous structure (see Eq 7 in methods). Since a typical liquid volume of a "sip" (~5–15 ml) is significantly larger than tongue surface volume, we assume the concentration in the liquid phase is constant (i.e., does not decrease due to diffusion into papilla structure).

We compared our simulated tongue surface concentration results with several experiments found in the literature. Fig 3A shows experimental time-intensity profiles when subjects sip and hold 450mM NaCl for 10s[7]. The positive correlation between the simulated averaged tongue surface concentration and the experimental intensity ratings was 0.99. Fig 3B shows peak perceived intensities for holding 150, 300, or 450mM NaCl[7] at 5s or 10s (blue triangles) or 100, 180, 320, or 360mM sucrose for 10s[8] (red dots; see also S2 Table). Overall, the tongue surface concentration buildup over time matches perceived intensity. Since diffusion dominates the transport process when fluid movement is limited, it is natural to ask whether the diffusivity of the stimuli affects the taste temporal profile. We compiled the diffusivities of sweet compounds used in literature[9] and found that they greatly correlate to the slope and time to reach peak perceived intensity and time lag of first perceiving taste (Fig 3C and 3D; see also S3 Table). The negative or positive correlations in Fig 3C and 3D indicates that a high diffusivity of a compound would lead to shorter time to reaching peak, with a steeper slope and shorter time lag to first perceive taste.

## Discussion

Compared to other sensory perceptions, human taste has significant delays, and it has long been speculated that early stimulus transport events that initiate taste perception may contribute to this temporal lag[6]. However, relatively little is known about these early events because of the complexity of stimulus transport within the human tongue papilla structure, which

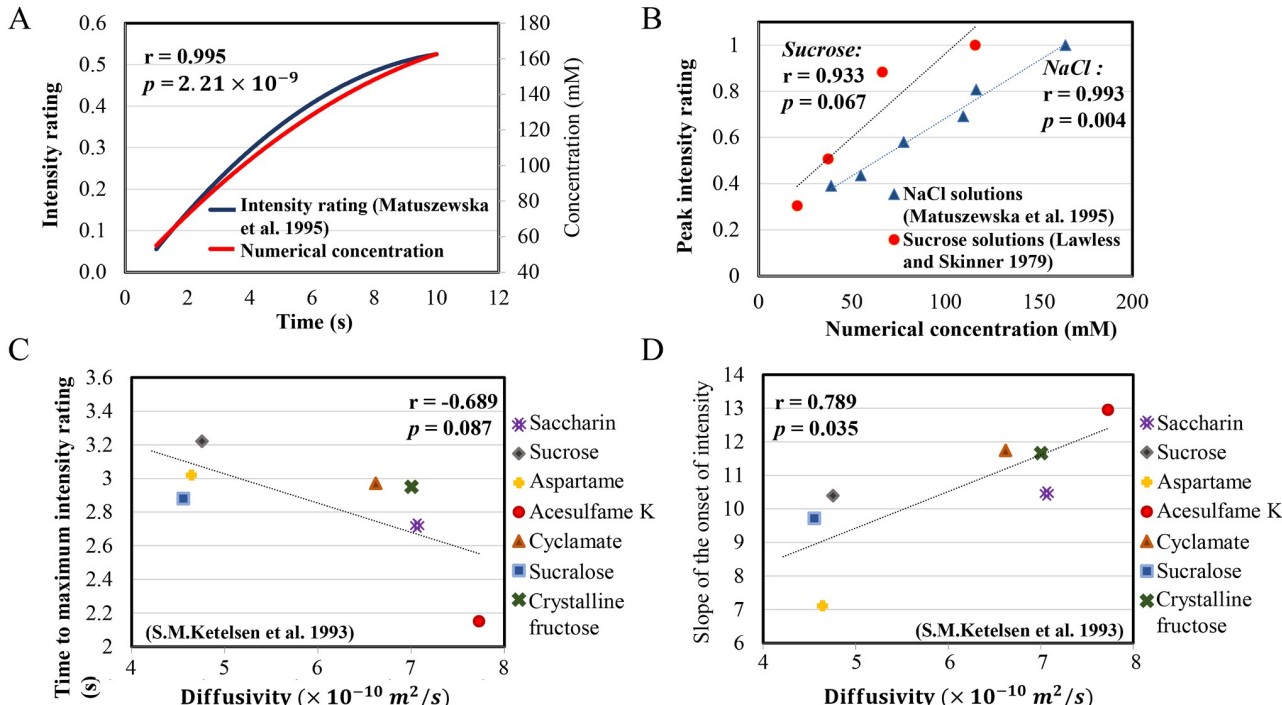

**Fig 3.** *In silico* **replication of sip-and-hold experiments. (A)** Time-intensity profiles of subjects sipping and holding 450mM NaCl for 10s (blue curve) vs. the simulated time-concentration profile of the stimuli within the tongue surface zone during the same time period (red curve). **(B)** Correlation of peaks of intensity perceptions against the simulated peak stimulus concentrations in the tongue surface zone for various setups: NaCl (blue triangles) at 150, 300, or 450mM, holding for 5s or 10s; and sucrose (red circles) at 100, 180, 320, or 360mM, holding for 10s. The peaks of intensity were normalized by the maximum value in each experiment. **(C,D)** Correlations of the diffusivities of a list of sweet compounds against literature-reported time **(C)** and slope **(D)** to reach peak perceived intensity (for details, see S3 Table).

requires enormous computational or experimental effort to completely capture its details. Therefore, we tried a novel approach by modeling the structure of the human tongue surface as a porous medium. By estimating key porous properties—permeability and porosity based on papilla histology—we simulated the transport of a range of sweet and salty stimuli in the region and validated its accuracy against experimental measured concentration profiles. Our *in silico* replication of Kelling and Halpern's temporal stimulus intensity experiments provided quantitative evidence that the accumulation of stimulus concentrations has significant delay (Fig 2B and 2C) and seems to match well with the temporal profile of experimentally perceived intensities, especially at earlier time points (Fig 2D and 2E). This confirms previous speculation that peripheral transport processes may have significant impact in human gustatory perceptions[6,7].

Our simulation also explained key differences between the tastant delivery experiment by Kelling and Halpern and vast sip-and-hold experiments in the literature. In the rapid-stimulus-pulse experiments of Kelling and Halpern the perceived intensity peaked in <2s, whereas in most sip-and-hold experiments the intensity rises much more slowly, in the range of >2-10s. We believe the reason is that the flow rate in Kelling and Halpern's experiments was quite high, with velocity reaching 0.8 m/s, such that stimulus transport was dominated by convection. The fluid velocity in Kelling and Halpern's experiments is high enough to actually penetrate into the porous media (see S1 Video). In contrast, fluid movement during sip and hold may be limited, and transport is likely dominated by diffusion processes. Returning to our

kelp forest analogy, when the sea is calm, the kelp can form a barrier to mass transport, but when sea current is strong, the current can penetrate into the kelp forest, and convective transport is far more effective than diffusion transport (the difference between Figs 2B and 2C vs 3A).

Our simulated sweet and salty stimuli concentrations match both types of experiments quite well and can explain the significant temporal difference between them. This is the key improvement of our approach over previous tongue surface models[3,18,26], where the tongue surface is traditionally modeled as a compartment that the fluid movement can convectively transport stimuli to the interface between tongue and liquid but can't penetrate through the interface. In our diffusion-dominant sip-and-hold simulations, we already assumed that the interface is at saturated concentration and that the convective transport in the fluid zone is 100% efficient, but the simulation results still fall significantly short of Kelling and Halpern's results. This indicates that the diffusion process alone, without accounting for fluid penetration into the tongue's surface, can't capture differences between two types of experimental set-ups. This also indicates that a range of fluid movement in the mouth may greatly impact taste perception, which needs to be taken into account in reviewing vast sip-and-hold experiments in the literature. For example, in some studies subjects are instructed to sip and hold still, while in others subjects are allowed to stir the fluid with the tongue or in the mouth. This could explain why the results between studies can be very different. This may also provide the mechanism to explain why higher-viscosity taste solutions have been reported to have reduced taste intensity[30–34]. Potentially higher viscosities would restrict fluid movement more.

In addition to fluid movement, does the diffusivity of stimuli affect taste perception? Diffusivity of a compound largely depends on its molecular volume as characterized by the classic 1905 Stokes–Einstein equation[35], with smaller sizes leading to higher diffusivities, higher diffusion rates, and quicker concentration increases. To test this hypothesis, we compiled data from the literature and showed that diffusivities for a list of sweet compounds greatly correlate with their measured taste temporal profile, with high diffusivity leading to shorter time to reach peak, with a steeper slope and shorter time lag to first perceive taste. This serendipitous finding further indicates that taste stimulus transport may significantly impact early taste perceptions. Thus, future psychophysics taste experiments need to consider not only the concentration of stimuli entering the oral cavity but also real-time concentration changes within the tongue surface. Other modalities of taste stimuli, such as bitter, umami, and sour compounds, also need to be investigated in the future.

Since taste buds are located in different regions of the papillae (e.g., near the apical surface of fungiform papillae vs. in deep groves between circumvallate papillae), one might question the concentration variations at different depths of the papilla layer. However, this spatial relationship is not preserved in a porous media approach. For example, when water soaks into a sponge, some deep pockets may fill quickly due to potentially easy access path, whereas some small pores even near the surface may be more difficult to penetrate due to it being well pocketed. We know the sponge will soak up more water with time, but we can't predict how fast each particular pocket or pore will be penetrated—this would require modeling every pocket structure in the sponge precisely, which negates the advantage of the porous-medium approach. Similarly, stimuli many not necessarily penetrate a taste pore near the surface before penetrating one deeper in the papilla structure. Rather, the overall rise in concentration might be a better indicator of the likelihood of stimuli penetrating taste pores in general.

In summary, we believe this study will have broad implications across many disciplines and fields—engineering, neurobiology, food science, and public health: 1) our study demonstrates the novelty of modeling the human tongue as a porous material to significantly simplify analytical and computational approaches, introducing well-established methodologies from

nanofiber engineering to a new biological application. 2) Our results indicate that peripheral transport processes may have a significant impact on human taste perceptions. Our results may inspire future biology, neuroscience, and behavior investigations that could have a significant impact on global public health, especially if we can modulate taste responses by adjusting transport processes, for example, speeding up and shortening the transport processes of sweet compounds to enhance the onset of sweet taste perception, achieving rapid sensory satisfaction while reducing sugar intake, or prolonging transport processes to delay the onset of bitter tastes, to improve the tolerance of bitter medicine. These questions may have significant scientific and biomedical impacts.

## Methods

### Porous media properties of the human tongue surface: Permeability and porosity

In this study, we considered the human tongue surface as a porous medium. Inspired by previous work[18,26], we simplified the papilla structure to a rod-like porous medium model (Fig 2A). The height of the rods was assumed to be equal to the thickness of the papilla layer, 0.3mm based on literature[18]. The anterior region of human tongue has mainly two types of papillae, fungiform and filiform, with the latter of greater density. As summarized in Table 1, we chose various distribution densities and diameters to cover the range of values reported in the literature[18,20,22,26].

Porosity is a key parameter of the porous medium and was determined by direct volume method[17]:

$$\varepsilon = \frac{V_s}{V_t}, \tag{1}$$

where $V_s$ the solid volume of the human tongue surface region and $V_t$ is the total volume of the region.

Permeability of porous medium depends on both its morphology and its porosity. We applied a published empirical formula, adapted for porous fiber[12,16]:

$$\frac{K_p}{r_f^2} = \frac{1}{8(1-\varepsilon)}\left(-\ln(1-\varepsilon) - 1.476 + 2(1-\varepsilon) - 1.774(1-\varepsilon)^2 + 4.076(1-\varepsilon)^3\right), \tag{2}$$

where $r_f$ is the average radius of papillae and $\varepsilon$ is the porosity of the predicted porous media.

### Taste stimulus transport in human tongue surface replicating Kelling and Halpern's experiment

We applied the two-domain approach[36] to *in silico* replicate the experimental setup. In the free-flow domain in the delivery tube, we considered the stimulus solution as a viscous, incompressible turbulent flow and solved the steady-state momentum by the well-established k–ω method. In the porous media domain (i.e., the tongue surface domain), we considered the flow as laminar and solved the momentum by Brinkman's equation[13,36]:

$$-\nabla p + \mu_e \nabla^2 \mathbf{u} - \frac{\mu}{K_p}\mathbf{u} = 0, \tag{3}$$

where $\mathbf{u} = (u, v, w)$ represents the velocity throughout the simulated domain; $\nabla p$ is the pressure gradient; and $\mu_e$ is the effective viscosity: as $\mu_e/\mu = 1/\varepsilon$ [37,38], $\mu$ is the viscosity of the stimulus solution in the free-flow domain. Note that in low-porosity environments the

Laplacian term in the Brinkman's equation is not valid. Therefore, we could only consider conditions for $\varepsilon \geq 0.6$. Generalization to smaller porosity values will be the subject of future investigations.

The stimulus concentration was described by the convection-diffusion equation[13,39] in porous media:

$$\varepsilon \frac{\partial c}{\partial t} = \varepsilon D \nabla^2 c - \mathbf{u} \cdot \nabla c, \tag{4}$$

where c is the concentration of the stimulus and D is the diffusivity of the stimulus in water. In the present study, we considered the diffusivity of NaCl in water to be $1.47 \times 10^{-9}$ m$^2$/s[40]; the diffusivity of NaSac to be $0.7 \times 10^{-9}$ m$^2$/s, the latter was estimated based on the well-known Wilke and Chang equation[41,42]:

$$D = \frac{7.4 \times 10^{-8} (\beta M)^{\frac{1}{2}} T}{\eta V^{0.6}}, \tag{5}$$

where $\beta$ is called the association parameter to define the effective molecular weight of the solvents with respect to the diffusion process; in water $\beta = 2.6$. $M$ is the molecular weight of the solvent, $V$ is the molar volume of solute at normal boiling point, $T$ is the temperature in Kelvin, and $\eta$ is viscosity of the solution.

## The method of numerical simulation

The numerical simulations of fluid-flow momentum and the stimulus concentration transport were performed using commercial computational fluid dynamics software ANSYS Fluent 19.2. Fig 4A shows the geometry and meshes of our model for replicating Kelling and Halpern's intensity experiment. A 4mm-diameter, 40mm-long polypropylene tube with a 10×5 mm opening was placed on the anterior dorsal surface of the human tongue. The solution was delivered at a constant 10 ml/s, where pulses of tastant (500mM NaCl or 2mM NaSac) were switched on and off by an inlet valve. A constant velocity of 0.8 m/s at the inlet that matches experimental flow rates was applied for the momentum calculation. The flow had a relatively high Reynolds number, about 3200 inside the delivery tube. We assumed left-right symmetry to reduce calculation time and used extremely fine mesh elements in the tongue section to ensure the best simulation results (Fig 4A and 4B). Fig 4C shows the mesh independency validation results using 500mM NaCl as stimulus. The results converge for meshes over 0.8 million. Thus, the 1.4-million-mesh model was adopted in present study. Non-slip boundary conditions were considered on the wall.

## The diffusion-dominant model representing sip-and-hold experiments

The diffusion-dominant transport during sip and hold was simulated by a 1-D diffusion equation in porous media[43,44]:

$$\frac{\partial c}{\partial t} = D_{eff} \frac{\partial^2 c}{\partial x^2}, \tag{6}$$

$D_{eff}$ is the effective diffusivity within the porous media. A commonly used estimation of effective diffusivity[43,44] is

$$D_{eff} = \frac{D \varepsilon}{\tau}, \tag{7}$$

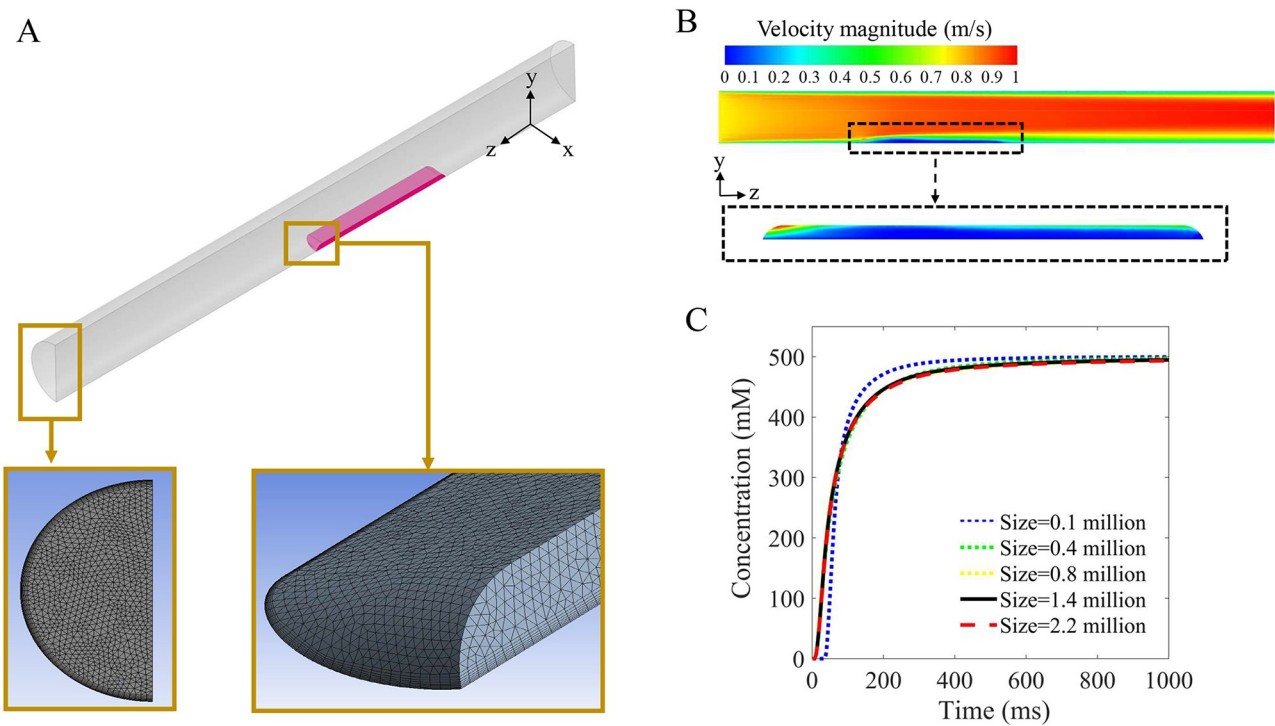

**Fig 4. Numerical simulation. (A)** The numerical model and mesh replicating Kelling and Halpern's experiment. **(B)** A longitudinal cross section of velocity contour plot at 10 ml/s. As shown in the inset, flow velocity in the porous tongue surface zone is not zero. See S1 Video for more details. **(C)** Grid independency verification based on simulation of 500mM NaCl in 1000ms. The results converge for meshes over 0.8 million.

where τ is the tortuosity of the porous material. Here, since our simulated porous media has a rod-like structure, we consider the tortuosity $\tau \sim 1$.

We considered the porous medium to have a thickness L = 0.3mm, which is the same as the papilla height. The boundary conditions and the corresponding analytical solution were as follows:

$$c(x, 0) = 0, \tag{8}$$

$$c(0, t) = 1, \tag{9}$$

$$\frac{\partial c}{\partial x}(L, 0) = 0, \tag{10}$$

The effective diffusivity depends on the structure of porous media. The analytical solution for Eqs 6–10 is

$$c = \sum_{n=1}^{\infty} \left[ \frac{-2}{\left(n - \frac{1}{2}\right)\pi} \right] \cdot e^{-\left[\frac{\left(n-\frac{1}{2}\right)}{L} \cdot \pi\right]^2 D_{eff} \cdot t} \cdot \sin\left[ \frac{\left(n - \frac{1}{2}\right)}{L} \cdot \pi \cdot x \right] + 1, \tag{11}$$

All the diffusivities of sweet compounds in water were calculated based on the Wilke and Chang equation (Eq 5). The detailed values are summarized in S3 Table.

## Supporting information

**S1 Fig. Top-surface view (top row) and cross-section view (bottom) of simulated NaCl concentration profile throughout the papilla layer at different time points, with stimulus of 500mM NaCl. (A)** Case 1: $K_p$ (permeability) = $1.18 \times 10^{-9}$ m$^2$, $\varepsilon$ (porosity) = 0.63; (**B**) Case 2: $K_p$ = $2.78 \times 10^{-9}$ m$^2$, $\varepsilon$ = 0.79; (**C**) Case 3: $K_p$ = $7.05 \times 10^{-9}$ m$^2$, $\varepsilon$ = 0.92. Parameters for each case are listed in Table 1.
(DOCX)

**S1 Table. Intensity perception ratings and simulated stimulus concentrations for different pulse durations of NaCl and NaSac stimuli.**
(DOCX)

**S2 Table. Peaks of intensity perceptions and simulated peak stimulus concentrations in the tongue surface zone for various setups.**
(DOCX)

**S3 Table. Estimated diffusivity of selected sweeteners and mean time-intensity parameters.**
(DOCX)

**S1 Video. The study procedures of *in silico* replication of Kelling and Halpern's experiment.**
(MP4)

## Author Contributions

**Conceptualization:** Kai Zhao.

**Data curation:** Zhenxing Wu, Kai Zhao.

**Formal analysis:** Zhenxing Wu, Kai Zhao.

**Funding acquisition:** Kai Zhao.

**Investigation:** Zhenxing Wu, Kai Zhao.

**Methodology:** Zhenxing Wu, Kai Zhao.

**Project administration:** Kai Zhao.

**Resources:** Zhenxing Wu, Kai Zhao.

**Software:** Zhenxing Wu.

**Supervision:** Kai Zhao.

**Validation:** Zhenxing Wu, Kai Zhao.

**Visualization:** Zhenxing Wu, Kai Zhao.

**Writing – original draft:** Zhenxing Wu, Kai Zhao.

**Writing – review & editing:** Zhenxing Wu, Kai Zhao.

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
