## [Decision Letter · Decision Letter 0]

11 Apr 2020

Dear Dr. zhao,

Thank you very much for submitting your manuscript "Taste of time: a porous-medium model for human tongue surface with implications for early taste perception" for consideration at PLOS Computational Biology. As with all papers reviewed by the journal, your manuscript was reviewed by members of the editorial board and by several independent reviewers. The reviewers appreciated the attention to an important topic. Based on the reviews, we are likely to accept this manuscript for publication, providing that you modify the manuscript according to the review recommendations.

Sincerely,

Samuel J. Gershman

Deputy Editor

PLOS Computational Biology

[LINK]

Reviewer's Responses to Questions

**Comments to the Authors:**

Reviewer #1: The paper from Wu and Yhao provides an original paper on modeling the surface of the human tongue

as a porous medium in order to investigate temporal taste profile. The work is highly innovative and covers a topic that is not enoght studied. It is well written and worth to be published.

I have just a minor issue that would need to be addressed. The authors discuss about taste in general but the model was applied only to NaCl, NaSac and other sweet compounds, that do not test any bitter, sour, umami compounds and do not cover all basic taste modalities, i.e. sweet, bitter, salty, sour, umami. I would suggest to re-write the title and re-phrase discussion and conclusion specifying that and softening conclusions about taste in general. Since 1983-86, publication years for the papers of Kelling and Halpern, much more is known about taste compounds and their complicated chemistry. So I think that it is important to frame the applicability of the model developed in this work to the compounds that were actually tested and discuss the possible applicability of this model for other compounds and all taste modalities as a perspective.

Reviewer #2: Dear Authors,

The manuscript "Taste of time: a porous-medium model for human tongue surface with implications for early taste perception" is interesting and with new data. Please find below my comments in details:

1. The aim of this study was clearly defined.

2. The claims of this study are partially novel.

3. The claims are properly placed in the context of the previous literature.

4. The results of this study are well described. The significance of the results of Yours study can have the influence on the future physiological research of the sense of taste.

5. The methods which were used in this study are correct. Authors attached the additional file for support of this study.

6. The manuscript contains the original data. Some of the parts of figures are supporting from the reference list.

7. The details of the methodology are sufficient to allow the experiment to be reproduced.

8. The manuscript is well organized.

9. When I was checking this manuscript I found the previously version of this manuscript on-line:

https://www.biorxiv.org/content/10.1101/780429v3

10. The nomenclature used in this manuscript is correct.

**Have all data underlying the figures and results presented in the manuscript been provided?**

Reviewer #1: Yes

Reviewer #2: Yes

PLOS authors have the option to publish the peer review history of their article (what does this mean?). If published, this will include your full peer review and any attached files.

Reviewer #1: No

Reviewer #2: No
---

## [Editor Report · Decision Letter 1]

20 Apr 2020

Dear Dr. zhao,

We are pleased to inform you that your manuscript 'Taste of time: a porous-medium model for human tongue surface with implications for early taste perception' has been provisionally accepted for publication in PLOS Computational Biology.

Best regards,

Samuel J. Gershman

Deputy Editor

PLOS Computational Biology

---

## [Editor Report · Acceptance letter]

19 May 2020

PCOMPBIOL-D-20-00270R1 

Taste of time: a porous-medium model for human tongue surface with implications for early taste perception

Dear Dr Zhao,

I am pleased to inform you that your manuscript has been formally accepted for publication in PLOS Computational Biology. Your manuscript is now with our production department and you will be notified of the publication date in due course.

With kind regards,

Sarah Hammond
